# Food and Medicinal Uses of Ancestral Andean Grains in the Districts of Quinua and Acos Vinchos (Ayacucho-Peru)

**Roberta Brita Anaya** [1], **Eusebio De La Cruz** [2], **Luz María Muñoz-Centeno** [3,*], **Reynán Cóndor** [1], **Roxana León** [4] **and Roxana Carhuaz** [1]

[1]    Faculty of Biological Sciences, National University of San Cristobal de Huamanga, Ayacucho 05001, Peru; roberta.anaya@unsch.edu.pe (R.B.A.); reynan.condor@unsch.edu.pe (R.C.); roxana.carhuaz@unsch.edu.pe (R.C.)
[2]    Chemical Engineering and Metallurgy, National University of San Cristobal de Huamanga, Ayacucho 05001, Peru; eusebio.delacruz@unsch.edu.pe
[3]    Department of Botany and Plant Physiology, University of Salamanca, 37008 Salamanca, Spain
[4]    Faculty of Health Sciences, National University of San Cristobal de Huamanga, Ayacucho 05001, Peru; roxana.leon@unsch.edu.pe
*    Correspondence: luzma@usal.es

**Abstract:** Andean grains are key elements in the construction of family production systems. These seeds speak of the history of a people, their customs and ancestral knowledge. The general objective of the work was to evaluate the food use, crop management and traditional knowledge about the medicinal use of ancestral Andean grains among the inhabitants of the districts of Quinua and Acos Vinchos (Ayacucho-Peru). Basic descriptive research, carried out by means of convenience sampling, the sample size determined by the Law of Diminishing Returns, after signing an informed consent form. Semi-structured individual interviews were applied to 96 informants. A total of 96.9% of the informants reported that they obtained *quinoa* grain from their own crops, and 24.0% obtained *achita* grain that they sowed directly on their land; no *cañihua* was cultivated. A total of 58.3% use *quinoa* and *achita* in their diet. The variability of the food use of ancestral grains, specifically *quinoa* and *achita*, constitute a natural source of vegetable protein of high nutritional value, which represents one of the main foods of the inhabitants of Quinua and Acos Vinchos. Traditional medicine derived from the ancestral knowledge of Andean grains is barely preserved, but this is not the case for other medicinal plants in the area, as this knowledge is still preserved.

**Keywords:** *Chenopodium quinoa* Willd (quinoa); *Amaranthus caudatus* L. (achita); *Chenopodium pallidicaule* Aellen (cañihua); ethnobotany; Andean grains; food uses; medicinal uses; edaphic resilience and crops

## 1. Introduction

Andean grains have a high value, as they are key elements in the construction of Andean family production systems. These seeds speak of the history of a people, their habits and resistance. As Arias states [1], defending Andean grains is like protecting real possibilities of an independence that defies the market and money. Currently, the relationship created so many years ago between the inhabitants of this area and their grains is being lost, resulting in the loss of this genetic material and the ancestral knowledge related to its management and uses. Plants are present in all areas of human activity, and ethnobotany combines the application of various disciplines to understand the relationship between a culture and the plant world that surrounds it [2]. It is a science that links botany and anthropology and also draws on other disciplines such as ecology, pharmacognosy, medicine, nutrition, agronomy, sociology, linguistics and history. The interaction between humans and plants is one of the aspects that indicates how a culture relates to the natural environment, and ethnobotany is therefore situated within ethnoecology. Ethnoecology approaches the study of traditional cultures not as obsolete systems but as a fraction of

society that possesses valuable resilient ecological wisdom. According to Aceituno's review, a distinction is made within ethnobotany between the cognitive stream, concerned with how humans perceive and classify plants, and the utilitarian stream, concerned with how they use and manage them. The former uses methods from the social sciences, while the latter uses an approach from the natural sciences [2].

The FAO, the Food and Agriculture Organization of the United Nations, considers quinoa (*Chenopodium quinoa* Willd.) to be one of the most promising foods for humanity, along with achita (*Amaranthus caudatus* L.) and cañihua (*Chenopodium pallidicaule* Aellen), not only because of its great beneficial properties and multiple uses, but also because it is considered an alternative to solve serious human nutrition problems, especially because of its ability to grow in adverse environments [3,4]. Food security has been affected by many factors, including the growing demand for basic agricultural commodities for production, the disappearance of traditional varieties that are highly resilient to climate change, the increase in the world's hungry population, inadequate food and nutrition policies in the country, and the misallocation of domestic and foreign aid, all of which have contributed to malnutrition among the most vulnerable populations and, consequently, to deficiencies in the integral development of the human being. The recent COVID-19 pandemic has exacerbated this situation. Therefore, if the world is to cope with pandemics without stopping food production and distribution, the cultivation of staple food crops, which are basically grain crops, will have to be increased. Crops that are drought, pest and disease-resistant, nutritionally superior, and health-promoting will have to be selected. These include *quinoa*, *achita* and *cañihua*. In addition, future grain crops must be produced in a profitable, sustainable and environmentally friendly way [5].

This ethnobotanical study has focused on the ancestral medicinal and nutritional knowledge of Andean grains, due to their great benefits as a protein and functional food that reduces the risk of various diseases. *Quinoa*, in particular, is considered one of the most important grains of the 21st century, sometimes referred to as a "superfood" It is high in protein, carbohydrates, polyunsaturated fatty acids, vitamins, fiber and minerals and is gluten-free. It also has high concentrations of bioactive compounds such as phenolic acids, flavonoids, bioactive peptides, saponins, phytosteroids and phytosterols. [6–9]. These bioactive compounds provide quinoa with anti-diabetic, antioxidant, anti-inflammatory, immunoregulatory, antimicrobial, anti-obesity, anti-cancer and heart-healthy properties [10,11].

The Andean region comprises one of the eight largest centers of cultivated plant domestication in the world, giving rise to one of the most genetically diverse and sustainable agricultural systems in the world. *Quinoa* is a good example of this, showing a great diversity of genotypes and wild progenitors in Peru and Bolivia; this crop groups around 3000 varieties. Currently, Peru is the world's leading producer of quinoa, surpassing Bolivia [6,12–14].

With the results of this research, we intend to highlight the great importance of these natural resources and their associated knowledge in order to recover their mass consumption in all social strata, especially in those most in need, and thus reduce the high levels of anemia and malnutrition in which a large part of the population of this area is immersed. Sustainable cultivation based on the local farmers' own grains and therefore on traditional varieties would provide alternative crops for self-sufficiency and trade that would make the population less dependent on introduced products with low nutritional levels, which a priori are cheaper, but which in the long run produce irreparable consequences such as chronic malnutrition in these Andean peoples [13,15].

Our research hypothesis is to study whether traditional knowledge on the use and management of Andean grains is still valid or has been lost due to globalization. The main aim of this research is to recover, evaluate and enhance local knowledge about ancestral Andean grains in the districts of Quinua and Acos Vinchos (Peru). To this end, the specific objectives of this study are (a) to collect data on the socio-economic characteristics of the informants; (b) to evaluate the information on the nutritional use of these grains; (c) to

determine the type of management of these crops; (d) to collect traditional knowledge on the medicinal use of these grains and assess its validity.

## 2. Methodology

### 2.1. Study Area

The study was carried out in the districts of Quinua (3280 m asl) and Acos Vinchos (2840 m asl) in the province of Huamanga, department of Ayacucho-Peru (Figure 1). The province of Huamanga is located in the Peruvian Andes and presents a very diverse physiography with a particular flora and vegetation. Pulgar [16] defined and described the existence of eight natural regions in Peru based on geographical location and natural indicator vegetation. Tapia [17] has added agronomic variables to this classification into natural regions and proposes a classification into agro-ecological zones, which he complements with local peasant knowledge, crops, varieties and cultivation practices. According to this zoning of agro-ecological zones in the Peruvian Andes, the study districts are in the Quechua region (between 2300 and 3500 m above sea level). The climate in this region is temperate, varying according to latitude. Temperatures can fluctuate between an average annual temperature of 11 to 16 °C, with maximums between 22 and 29 °C and minimums between 7 and 4 °C during the winter (May to August); humidity rates are between 500 to 1200 mm of precipitation, increasing from south to north. These conditions make it possible to differentiate the Quechua region into three zones: arid, semi-arid and semi-humid. In 2019, the year of sampling, the average rainfall in the study area was 893 mm; the rainy season covered about 7 months (October–April), with an average total accumulation of 74 mm [14].

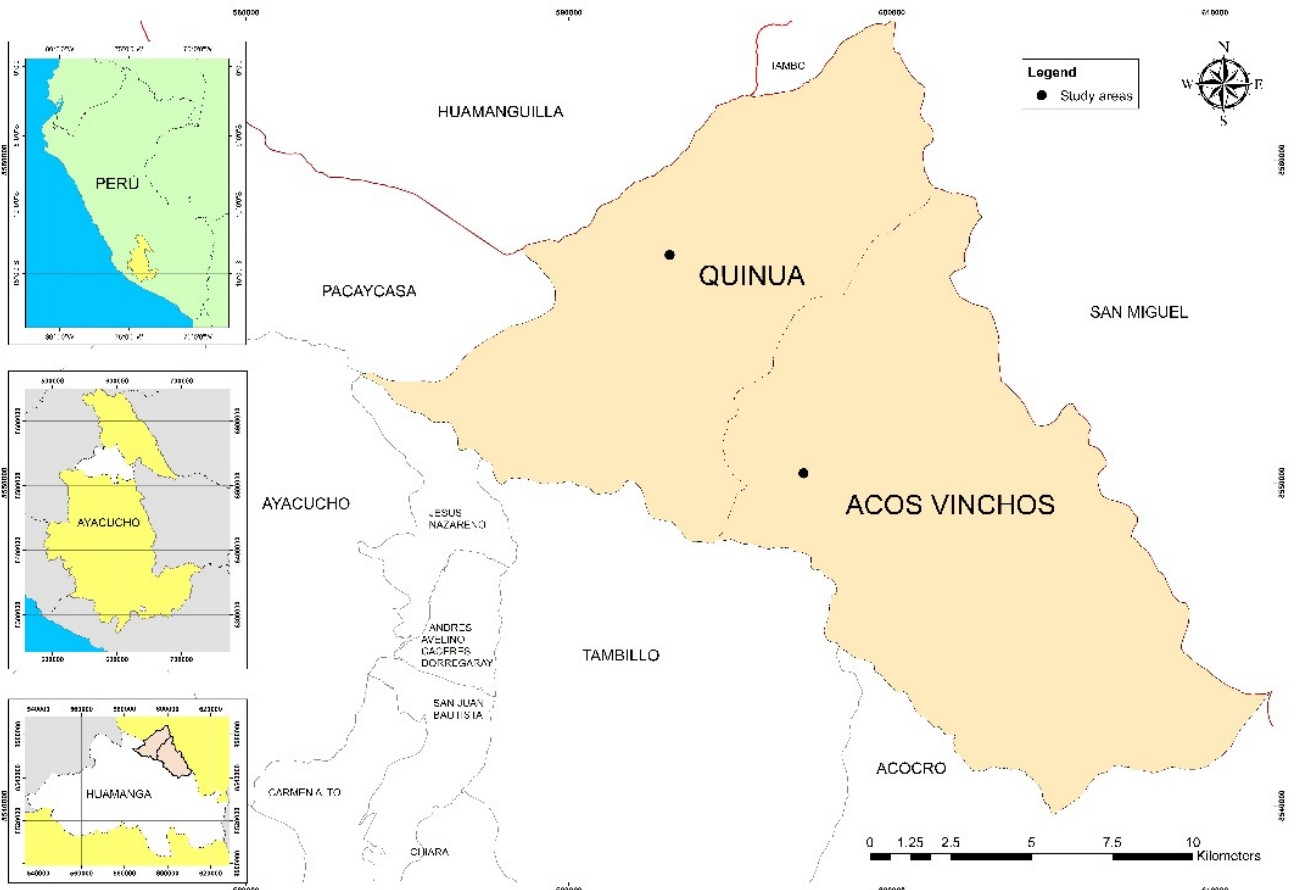

**Figure 1.** Location map of the study areas: Quinua and Acos Vinchos, province of Huamanga, Region Ayacucho-Peru.

*2.2. Ethnobotanical Data Collection*

Ethnobotanical fieldwork was conducted between May and June 2019 in the urban and surrounding areas of the districts of Quinua and Acos Vinchos. We interviewed 96 informants (40 women and 56 men) selected purposively, looking for "experts" within the local population. [18,19]. By "local population" we mean people who have lived in the area since birth or who migrated more than 20 years ago. By "experts" we mean people who have kept part of the cultural richness related to plants in their memories or customs [20]. First of all, older people were interviewed because they are the ones who have the most ancestral knowledge about the use of plants in their area [21]. Informant age ranged from 27 to 75 years old, with a majority being between 45 and 60 years old.

The sample size is determined by the Law of Diminishing Returns and is therefore not defined in advance [22]. The surveys were conducted in Spanish and in Quechua (the native language), using previously validated questionnaires in which the same questions were always asked about these Andean grains. The basic interview was a one-to-one meeting.

After direct and participatory observation, information was collected through semi-structured individual interviews and field interviews. In the semi-structured interviews, the informants were conversed within a relaxed manner so that they felt comfortable and could transmit their knowledge in a simple and sincere way. Beneath this apparent simplicity, it is necessary to control how the interview is conducted, how the questions are constructed and presented, and how responses are recorded [23]. For this purpose, the previously validated questionnaire was used. Emphasis was placed on recording whether the uses, practices or beliefs are traditional, whether the informant has ever put them into practice or has only heard about them, and whether the use of these plants is still in use today, in order to record ancestral knowledge and study changes in their use and management. In addition to interviews in the urban area, interviews were also carried out in the houses around the fields and in the countryside.

Plants and seeds were collected in situ. The vouchers collected during the walks were dried and stored in the Herbarium Huamangensis of the Faculty of Biological Sciences of the National University of San Cristóbal de Huamanga (Ayacucho-Perú) and the seeds in the Biochemistry Laboratory of the National University of San Cristóbal de Huamanga [22,24]. The species were determined by Blga. Laura Aucasime Medina.

*2.3. Data Analysis*

The ethnobotanical information obtained in the surveys carried out with the inhabitants of Quinua and Acos Vinchos was organized in a Microsoft Office Excel 2019 database. The data analysis was carried out by obtaining descriptive statistics. Frequencies and percentages were used to elaborate statistical tables and graphs. Statistical processing and analysis were carried out in R version 3.6.3 [25].

**3. Results**

The results of the interviews were grouped into six frequency tables based on the questions asked about socio-economic profile of the surveyed villagers (Tables 1 and 2), food use of Andean grains (Table 3), agronomic management of Andean grains (Table 4), medicinal use of Andean grains (Table 5) and health care (Table 6).

With regard to the socio-economic profile (Table 1), the surveyed inhabitants of Quinua cover an age range between 38 and 65 years, with a mean of 53 years, SD of 6.74; most of them correspond to the age group between 45 and 60 years with 71.4%, and those who participated in greater proportion were men with a 20% difference. Regarding the level of education, 71% are between those with no education and those with only primary education; however, 57% practice agriculture; 21.4% are also housewives. A total of 97.6% are natives and live in Quinua, 83.4% of whom are between 45 and 65 years old, i.e., adults and older people who still preserve their ancestral knowledge.

**Table 1.** Socio-economic profile of the surveyed villagers in *Quinoa*.

| Variable | Frequency (*n* = 42) | % |
|---|---|---|
| **Age** (min = 38, max = 65, mean = 52.93, SD = 6.74) | | |
| <45 years | 5 | 11.9 |
| 45–60 years | 30 | 71.4 |
| >60 years | 7 | 16.7 |
| **Gender** | | |
| Female | 17 | 40.5 |
| Male | 25 | 59.5 |
| **Education level** | | |
| Higher Education | 1 | 2.4 |
| Secondary school | 8 | 19.0 |
| Basic education | 14 | 33.3 |
| No schooling | 16 | 38.1 |
| No answer | 3 | 7.1 |
| **Present occupation** | | |
| Merchant | 3 | 7.1 |
| Farmer | 24 | 57.1 |
| Housewife | 2 | 4.8 |
| Civil servant and farmer | 1 | 2.4 |
| Farmer and housewife | 9 | 21.4 |
| No answer | 3 | 7.1 |
| **Birth place** | | |
| Lima | 1 | 2.4 |
| Quinua | 41 | 97.6 |
| **Living region** | | |
| Quinua | 41 | 97.6 |
| Quinua-Chihuanpampa | 1 | 2.4 |
| **Years of residence** (min = 15, máx = 65, mean = 51.21, SD = 9.89) | | |
| <45 years | 7 | 16.7 |
| 45–60 years | 28 | 66.7 |
| >60 years | 7 | 16.7 |

Table 2 shows the socio-economic profile of the surveyed inhabitants of the district of Acos Vinchos, with an age range between 27 and 75 years, with an average age of 55 years, SD of 9.25, the largest age group being between 45 and 60 years with 63%, followed by older adults with 24.1%, and those who participated in the highest proportion were men with 59.3%, followed by women with 37%. Regarding the level of education, 90.8% are among those who have no education (51.9%) and the difference is that they have only primary education (38.9%); 61.1% of them are farmers, and 27.8% of them are housewives and traders. A total of 90.7% are locals from the area, and almost 100% live in Acos Vinchos and Huinchupata (district town). As shown in Table 2, it is the older adults who participated most in the interviews.

Regarding the food use of Andean grains (Table 3), 58.3% use *quinoa* and achita, and only 41.7% use *quinoa*. Paradoxically, in Acos Vinchos, they use more *quinoa* (66.7%) than in the district of Quinua (9.5%). The vast majority (87.5%) of *quinoa* and *achita* grains are used for food, and only 12.5% also use the tender leaves of *quinoa*. In both districts, these grains are consumed almost entirely (96.9%) at breakfast, as soups and seconds. A total of 66.7% cultivate only for consumption and 31.3% for commercialization. A total of 65.6% say they know which plants are more nutritious, compared to 28.1% who say they do not know.

Based on the results and in situ verification of Andean grain crops, most of the inhabitants grow *quinoa* for their own consumption and trade, and to a lesser extent *achita*. No cultivation of *cañihua* was found, confirming that this species thrives at higher altitudes than the areas visited, as it is typical of high altitudes such as Puno, where it reaches altitudes of around 3800 m above sea level [4,16].

**Table 2.** Socio-economic profile of the inhabitants surveyed in Acos Vinchos.

| Variable | Frequency ($n$ = 54) | % |
|---|---|---|
| **Age** (min = 27, max = 75, mean = 53.49, SD = 9.25) | | |
| <45 years | 6 | 11.1 |
| 45–60 years | 34 | 63.0 |
| >60 years | 13 | 24.1 |
| No answer | 1 | 1.9 |
| **Gender** | | |
| Female | 20 | 37.0 |
| Male | 32 | 59.3 |
| No answer | 2 | 3.7 |
| **Education level** | | |
| Secondary school | 3 | 5.6 |
| Basic education | 21 | 38.9 |
| No schooling | 28 | 51.9 |
| No answer | 2 | 3.7 |
| **Present occupation** | | |
| Merchant | 1 | 1.9 |
| Farmer | 33 | 61.1 |
| Housewife | 4 | 7.4 |
| Farmer and housewife | 12 | 22.2 |
| Merchant and farmer | 3 | 5.6 |
| No answer | 1 | 1.9 |
| **Birth place** | | |
| Acosvinchos | 49 | 90.7 |
| Ayacucho | 1 | 1.9 |
| Chaupirara | 1 | 1.9 |
| Chaupirara-Acosvinchos | 1 | 1.9 |
| Huinchupata | 1 | 1.9 |
| Lima | 1 | 1.9 |
| **Living region** | | |
| Acosvinchos | 53 | 98.1 |
| Huinchupata | 1 | 1.9 |
| **Years of residence** (min = 47, max = 70.5, mean = 52.31, SD = 10.67) | | |
| <45 years | 7 | 13.0 |
| 45–60 years | 32 | 59.3 |
| >60 years | 10 | 18.5 |
| No answer | 5 | 9.3 |

The results on the crop management of Andean grains by the inhabitants of Quinua and Acos Vinchos are shown in Table 4. A total of 96.9% of the informants' report obtaining *quinoa* grain from their own crops and 24.0% from the *achita* grain that they sow directly on their land. *Cañihua* is not cultivated in these districts because of the altitude required for its optimal development, given that it is a species native to the circum-lacustrine zone of Lake Titicaca, shared between Bolivia and Peru in the altiplano region, mainly at altitudes above 3800 m [16]. In terms of production, 39.6% of the informants say that the yield of the *quinoa* crop is good, and 56.3% that it is fair; in the case of *achita*, 6.3% say it is good and 18.8% say it is fair. A total of 93.8% of the informants use their own seeds for replanting.

Concerning the products used for the treatment and prevention of plant diseases caused by fungi and other pathogens, 85.4% of informants use chemical products and 10.4% use homemade products, mainly based on a combination of cupric sulphate and hydrated lime, dissolved separately in water in nonmetallic containers and without heating the ingredients.

**Table 3.** Food use and production of ancestral Andean grains in the districts of Quinua and Acos Vinchos. Ayacucho-Peru.

| | | Quinua | | Acos Vinchos | |
|---|---|---|---|---|---|
| | | **Frequency** | **%** | **Frequency** | **%** |
| Types of ancient Andean grains used for food | *Quinoa* | 4 | 9.5 | 36 | 66.7 |
| | *Quinoa* y *Achita* | 38 | 90.5 | 18 | 33.3 |
| | Total | 42 | 100.0 | 54 | 100.0 |
| Parts of the plant used for food | Grain | 36 | 85.7 | 48 | 88.9 |
| | Grain and young leaves | 6 | 14.3 | 6 | 11.1 |
| | Total | 42 | 100.0 | 54 | 100.0 |
| Form of consumption | Do not know | 1 | 2.4 | 2 | 3.7 |
| | Breakfast, main course and soups | 41 | 97.6 | 52 | 96.3 |
| | Total | 42 | 100.0 | 54 | 100.0 |
| Production for consumption only | Yes | 28 | 66.7 | 36 | 66.7 |
| | No | 11 | 26.2 | 17 | 31.5 |
| | Do not know | 3 | 7.1 | 1 | 1.9 |
| | Total | 42 | 100.0 | 54 | 100.0 |
| Production for trade | Yes | 11 | 26.2 | 19 | 35.2 |
| | No | 27 | 64.3 | 35 | 64.8 |
| | Do not know | 4 | 9.5 | 0 | 0 |
| | Total | 42 | 100.0 | 54 | 100 |
| They know which plants are the most nutritious | Yes | 34 | 81.0 | 29 | 53.7 |
| | No | 6 | 14.3 | 21 | 38.9 |
| | Do not know | 2 | 4.8 | 4 | 7.4 |
| | Total | 42 | 100.0 | 54 | 100.0 |

**Table 4.** Crop management of ancestral Andean grains in the districts of Quinua and Acos Vinchos. Ayacucho-Peru.

| | | Quinua | | Acos Vinchos | |
|---|---|---|---|---|---|
| | | **Frequency** | **%** | **Frequency** | **%** |
| Informants who crop quinoa | Sowed | 39 | 92.9 | 54 | 100.0 |
| | No answer | 3 | 7.1 | 0 | 0.0 |
| | Total | 42 | 100.0 | 54 | 100 |
| Informants who crop achita | Sowed | 8 | 19.0 | 15 | 27.8 |
| | Do not know | 34 | 81.0 | 39 | 72.2 |
| | Total | 42 | 100 | 54 | 100 |
| *Quinoa* production | Good | 18 | 42.9 | 20 | 37.0 |
| | Fair | 20 | 47.6 | 34 | 63.0 |
| | Do not know | 4 | 9.5 | 0 | 0.0 |
| | Total | 42 | 100.0 | 54 | 100.0 |
| *Achita* production | Good | 4 | 9.5 | 2 | 3.7 |
| | Fair | 5 | 11.9 | 13 | 24.1 |
| | Do not know | 33 | 78.6 | 39 | 72.2 |
| | Total | 42 | 100.0 | 54 | 100.0 |
| Origin of seeds for cultivation | Do not know | 2 | 4.8 | 1 | 1.9 |
| | Purchased | 2 | 4.8 | 1 | 1.9 |
| | Home-grown | 38 | 90.5 | 52 | 96.3 |
| | Total | 42 | 100.0 | 54 | 100.0 |
| Substances used to treat plant diseases | Chemicals | 37 | 88.1 | 45 | 83.3 |
| | Homemade products | 2 | 4.8 | 8 | 14.8 |
| | Do not know | 3 | 7.1 | 1 | 1.9 |
| | Total | 42 | 100.0 | 54 | 100.0 |

Dietary consumption of these Andean grains reduces the risk of suffering from certain diseases, thanks to their nutritional and functional characteristics. For example, quinoa has anti-diabetic, antioxidant, anti-inflammatory, immunoregulatory and anticarcinogenic properties, among others [10]. This is the reason why we wanted to complete our research by compiling the traditional knowledge of the medicinal uses of Andean grains of the inhabitants of the area (Table 5). The results show that only 13.5% recall any medicinal use of *quinoa*. A total of 76.9% use *quinoa* as a purgative and 23% for colic. The parts of the plant used for medicinal purposes are the grains (46.2%), the leaves (30.8%) and a mixture of leaves and grains (23.1%), which are harvested when the plant is ripe. It is administered as a wash (38.5%), as an infusion (30.8%), and a mixture of both, infusion and wash (23.1%). The interviewees do not remember how these infusions and washes are prepared and administered, nor do they provide information on the proportions (quantity of leaves or grains) to be included in the preparation, the cooking or maceration time, the dosage and frequency of administration. A total of 84.6% indicated that mixing *quinoa* with other plants could cause adverse effects, and 84.6% also stated that at high doses it could be toxic, without mentioning the reasons. Only 15.4% indicated that the medicinal use of *quinoa* is maintained, while 46.2% confirmed that it is disappearing.

**Table 5.** Medicinal use of ancestral Andean grains in the districts of Quinua and Acos Vinchos. Ayacucho-Peru.

| | | Quinua | | Acos Vinchos | |
|---|---|---|---|---|---|
| | | Frequency | % | Frequency | % |
| Informants who use quinoa and achita as medicinal plants | Yes | 7 | 16.7 | 6 | 11.1 |
| | No | 35 | 83.3 | 48 | 88.9 |
| | Total | 42 | 100.0 | 54 | 100.0 |
| Ailments for which quinoa is used | Purgante | 7 | 100.0 | 3 | 50.0 |
| | Colics-purgante | 0 | 0.0 | 3 | 50.0 |
| | Total | 7 | 100 | 6 | 100.0 |
| Part of the quinoa plant used | Grain | 3 | 42.9 | 3 | 50.0 |
| | Leaves | 3 | 42.9 | 1 | 16.7 |
| | Leaves and grains | 1 | 14.3 | 2 | 33.3 |
| | Total | 7 | 100.0 | 6 | 100.0 |
| Forms of administration of quinoa | Infusion | 3 | 42.9 | 1 | 16.7 |
| | Infusion and wash | 1 | 14.3 | 2 | 33.3 |
| | Wash | 3 | 42.9 | 2 | 33.3 |
| | Do not know | 0 | 0.0 | 1 | 16.7 |
| | Total | 7 | 100.0 | 6 | 100.0 |
| Adverse effects of quinoa mixed with other plants | Yes | 5 | 71.4 | 6 | 100.0 |
| | No | 2 | 28.6 | 0 | 0.0 |
| | Total | 7 | 100.0 | 6 | 100 |
| Quinoa toxicity in high doses | Yes | 6 | 85.7 | 5 | 83.3 |
| | No | 1 | 14.3 | 1 | 16.7 |
| | Total | 7 | 100.0 | 6 | 100.0 |
| Frequency of medicinal use of quinoa | Common use | 1 | 14.3 | 1 | 16.7 |
| | Disappearing | 5 | 71.4 | 1 | 16.7 |
| | Do not know | 1 | 14.3 | 4 | 66.7 |
| | Total | 7 | 100.0 | 6 | 100.0 |

**Table 6.** Health care and use of medicinal plants in the districts of Quinua and Acos Vinchos. Ayacucho-Peru.

| | | Quinua | | Acos Vinchos | |
|---|---|---|---|---|---|
| | | Frequency | % | Frequency | % |
| Places and practices carried out for health care | Health center | 5 | 11.9 | 5 | 9.3 |
| | Healer and use medicinal plants | 9 | 21.4 | 2 | 3.1 |
| | Health centre, healer and use medicinal plants | 28 | 66.7 | 47 | 87.0 |
| | Total | 42 | 100 | 54 | 100 |
| Places where medicinal plants are obtained | Purchased | 3 | 7.1 | 3 | 5.6 |
| | Collected from the field | 15 | 35.7 | 20 | 37.0 |
| | Grown in the orchard | 11 | 26.2 | 8 | 14.8 |
| | Collected from the field and orchard | 13 | 31.0 | 22 | 40.7 |
| | Do not use | 0 | 0.0 | 1 | 1.9 |
| | Total | 42 | 100.0 | 54 | 100.0 |
| The use of medicinal plants is common in the family | Yes | 39 | 92.9 | 49 | 90.7 |
| | Does not use | 3 | 7.1 | 5 | 9.3 |
| | Total | 42 | 100.0 | 54 | 100.0 |
| The use of medicinal plants is recommended | Yes | 39 | 92.9 | 54 | 100.0 |
| | Do not know | 3 | 7.1 | 0 | 0.0 |
| | Total | 42 | 100.0 | 54 | 100 |
| Informants who know healers in their community | Yes | 22 | 52.4 | 15 | 27.8 |
| | No | 17 | 40.5 | 39 | 72.2 |
| | Do not know | 3 | 7.1 | 0 | 0 |
| | Total | 42 | 100.0 | 54 | 100.0 |

Figure 2 shows that those interviewed between 45 and 60 years of age have the most knowledge about the medicinal use of these grains. A total of 11.9% of those were interviewed in Quinoa, and 7.5% in Acos Vinchos. This traditional knowledge is scarcely preserved among adults over 45 years of age, but it has been alarmingly lost among those under 45 years of age.

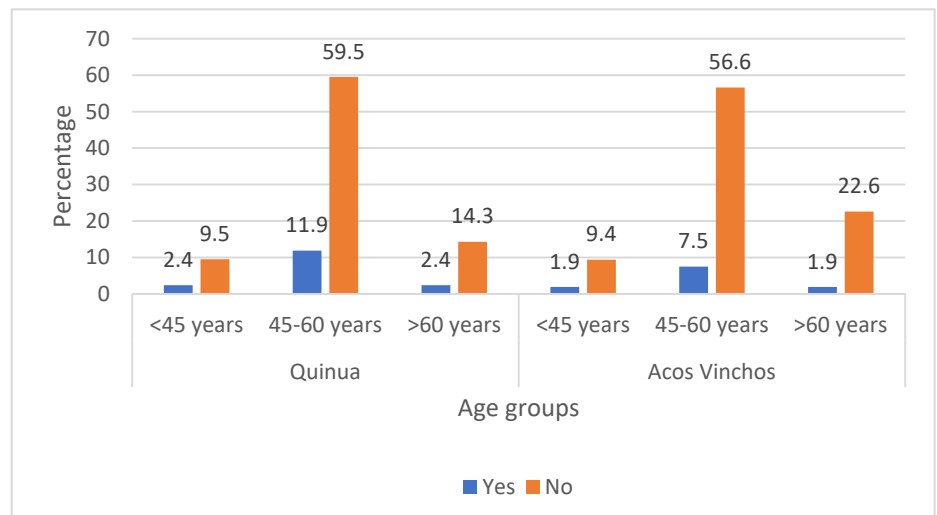

**Figure 2.** Response to the use of quinoa and/or cañihua plants as medicinal plant according to age groups.

While we asked about the traditional medicinal uses of Andean grains, we were also interested in the use of other medicinal plants and health care (Table 6). A total of 66.7% of the villagers interviewed from Quinua go to health center, use medicinal plants and visit the healer, and that number is 87.0% in Acos Vinchos. A total of 52.4% of those interviewed

in Quinua know a healer, and in Acos Vinchos, that number is 27.8%. They recognize the healers as having traditional knowledge about the medicinal use of plants.

Figure 3 shows that informants in both districts go to health centers but also use medicinal plants and visit healers to satisfy their primary health care needs. It is noteworthy that in the district of Quinua, there are still informants older than 45 years who do not go to the health center and instead take care of their health by using medicinal plants and visiting the healer; in Acosvinchos, the proportion of informants is much lower, and it is women older than 60 years who carry out these practices (use of medicinal plants and visiting the healers).

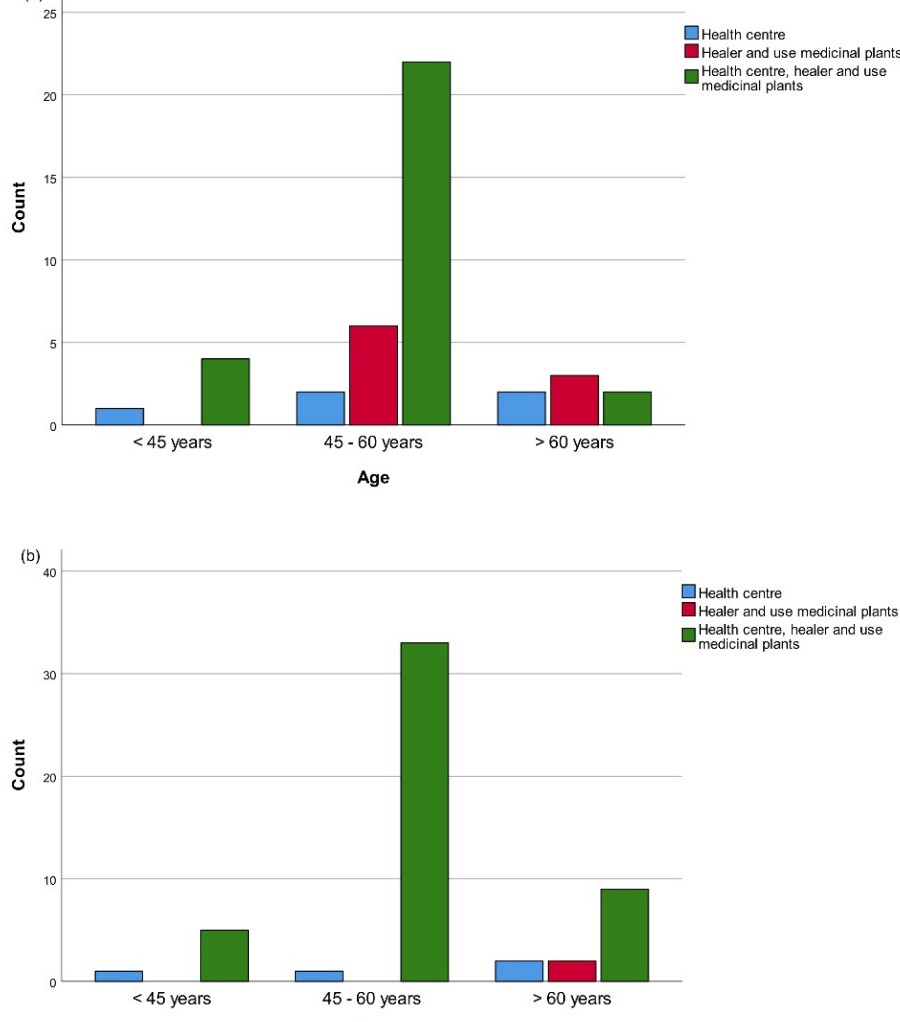

**Figure 3.** Places where respondents of Quinua (**a**) and Acos Vinchos (**b**) go when they become sick, according to age groups.

Figure 4 shows how the inhabitants of the districts studied obtain medicinal plants, according to gender. Men in Quinua (Figure 4a) mostly report that they collect them from the field, as well as from the field and the orchard. Very few people buy the plants to be used for medicinal purposes. In the case of women, the trend is similar, but to a lesser extent in all cases, except for purchasing. In Acos Vinchos (Figure 4b), the majority of men report that they collect them from the field and the orchard; a smaller proportion collect only from the field, and few grow them in the orchard. More women go to the fields to collect medicinal plants, although they also grow them in the orchard and buy them.

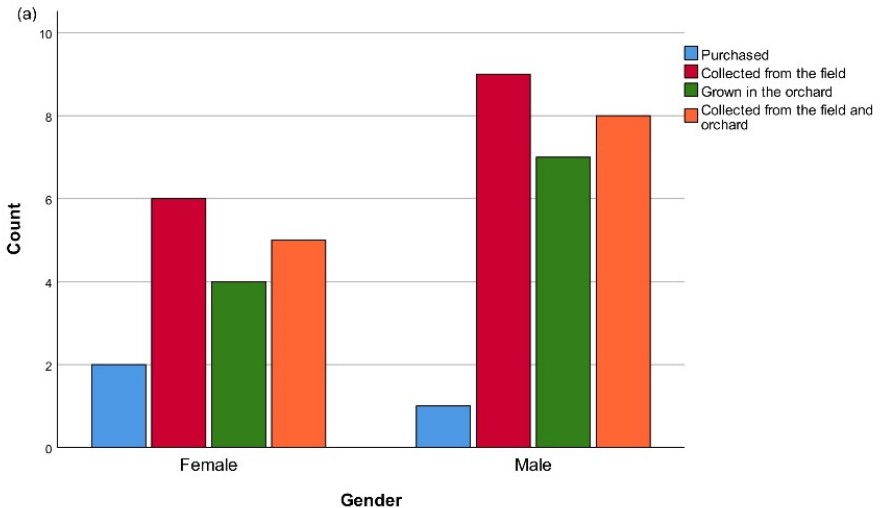

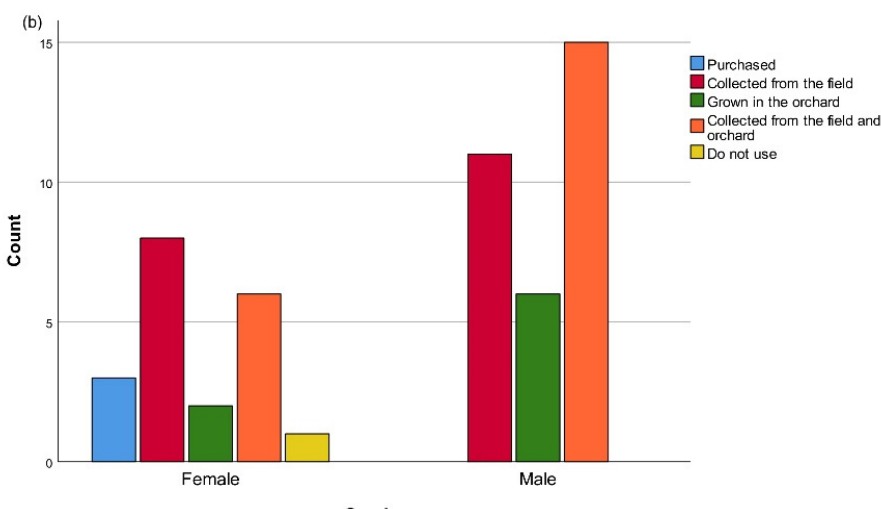

**Figure 4.** How the inhabitants of Quinua (**a**) and Acos Vinchos (**b**) obtain medicinal or aromatic plants according to gender.

## 4. Discussion

Peru is the world's largest producer of Quinoa, one of the most promising crops of the future [5,14]. It is important to know, evaluate and recover the knowledge that the villagers have about these ancestral Andean grains because this will indicate their degree of information about the benefits and advantages of cultivating these grains. This study has evaluated this knowledge with the collaboration of 96 informants, 40 women and 56 men, between 27 and 75 years of age, of different socio-economic characteristics and who were very participative. According to the information collected in Tables 1 and 2 and looking at the socio-economic profile of the two districts studied, it should be noted that, despite the low levels of education, the villagers have the practical management of their crops and have provided us with the required information about the use and management of these grains. Thanks are due to all participants for their generosity in sharing their knowledge with us.

It is important to remember that the Andes are the center of domestication of many cultivated species and that indigenous communities maintain native varieties. The characteristic crop of this agro-ecological region is maize, accompanied by cucurbits (*calabaza*, *caihua*, *zapallo*) and Andean grains (*quinoa*, *achita and cañihua*). *Quinoa* cultivation in the inter-Andean valleys of the Peruvian highlands was mostly carried out in a traditional system,

characterized by the sowing of quinoa in very small plots, interspersed with other species and on the edge of fields of other crops. Cultivation was mainly for self-consumption with native varieties, traditional soil preparation techniques, use of organic matter and their own pest control products, which gave them independence from external inputs. From the 2000s onwards, the growing demand in the national and international market and the high prices of quinoa had an impact on quinoa cultivation, resulting in an increase in the cultivation area and a change in the management of these crops [26–29]. The results obtained (Table 3) indicate that the most commonly used Andean grains in the districts studied are *quinoa* and *achita*, with more than half of the population interviewed, specifically 58.3%, consuming *quinoa* or *achita*. These grains are consumed in different ways and form part of the daily diet, a fact that helps to reduce the nutritional problems of the rural population, as these products are characterized by their high biological value proteins, as well as carbohydrates, polyunsaturated fatty acids, fiber, vitamins and minerals with antioxidant properties. It is considered a nutraceutical or functional food, with a high protein content (15.7% to 18.8%) and a significant proportion of essential amino acids, including lysine (7.1%), an amino acid that is scarce in plant-based foods and forms part of the human brain [6–9,30–32].

A total of 66.7% of the informants cultivate quinoa for their own consumption and, to a lesser extent, 31.3% for commercialization. Many farmers do not have adequate technological support, in addition to the problems of pests and drought, but the cultivation of *quinoa* and *achita*, even with these problems, is a profitable activity and is considered a good alternative for cultivation in the region [33]. The sowing of *quinoa* and *achita* has increased throughout the region, which is in line with Ministry of Agriculture (MINAGRI) information; thus, at the national level, the *quinoa* sowing area for the 2019–2020 agricultural campaign was 67,777 ha, slightly higher than the 2018–2019 campaign of 66,388 ha; in the Ayacucho region, 14,439 ha were sown in the 2018–2019 agricultural campaign, slightly lower than the 2017–2018 agricultural campaign in which 14,617 ha were sown [14]. Quinoa production is of economic and social importance for many Peruvian regions above 3000 m above sea level because there are not many other cultivation options. One example is the region of Puno, where this crop is the livelihood of approximately 100,000 rural families. This is one of the poorest regions in Peru with a poverty rate of 24.2% [34].

*Quinoa* has been selected as one of the crops destined to offer food security in the 21st century, due to its high capacity to withstand extreme environmental conditions and its qualities as a functional food. In addition, *quinoa* has gained space in international consumer markets in the last decade, which opens up economic opportunities for Andean producers [3,5,33,35–37].

The orchards are the best preserved agro-ecosystem in Peru, representing a biological and cultural refuge. Before 1960, the diversification of land use, cultivated varieties and the use of wild plants was a survival strategy. Cultural identity is one of the reasons for maintaining certain traditional uses, such as the collection of wild plants or the cultivation of orchards and traditional varieties [2].

In this perspective, our study shows a high percentage of informants (93.8%) who use their own seeds for replanting, thus becoming independent from commercially available seeds and maintaining local varieties, which is very positive. Regarding the adapted local varieties of *quinoa*, the most common and one of the most commercialized in Peru is the Blanca Junín, which has two types, Blanca and Rosada, and is typical of the central region. Other varieties are as follows: Hualhuas, obtained by the Universidad Nacional del Centro de Huancayo (white grains); Amarilla Maranganí, a variety from Cusco (Peru), with the characteristic of having a high saponin content; Roja Pasankalla, from the district of Ácora, province of Puno (red grains); Negra, INIA (2013) composed of 13 accessions, commonly known as "Quytu jiwras", which have their origin in the accessions that were collected from the localities of Caritamay, district of Ácora, province of Puno. The latter variety is also resistant to mildew [38,39]. In the Ayacucho Region, different varieties of quinoa (Amarilla Marangani, Illpa INIA, INIA Salcedo, Altiplano, Hualhuas, Rosada Junín, Huancayo, INIA 433, Pasankalla, Negra Collana, Amarilla Sacaca and Blanca Junín) from

different areas were evaluated and adapted in the farming community of San Antonio de Manallasacc in the district of Chiara, province of Huamanga and Ayacucho Region, Peru. The INIA 433 variety achieved the highest yield with 4.72 mt per hectare and 14.3% protein. The highest protein variety was Amarilla Marangani with 16% protein. Therefore, INIA 433 and Blanca Junín are the recommended varieties for this area because they have the best yields per hectare: 4.72 mt and 4.62 mt respectively [40].

With regard to the *achita* crop, the most common and most commercial is the variety of the farmer Oscar Blanco, (Ayacucho-INIA), with white grains. It responds to the agro-ecological conditions of the Ayacucho region in terms of health, production and grain quality. The *achita* production areas in Peru are mainly located in inter-Andean valleys with large climatic fluctuations that put the food security of small producers at risk. Furthermore, production technology is still traditional, sowing, field work and harvesting is manual, and mechanization is only used during field preparation and threshing, which is carried out with stationary threshing machines [4].

The use of agrochemicals is widespread in the care of *quinoa* and *achita* crops in the municipalities studied. A total of 85.4% of the interviewed villagers use them, with a small percentage (10.4%) using home-made products. These homemade products are used for the prevention and control of diseases caused by fungi and other pathogens and are composed of a combination of cupric sulphate and hydrated lime, dissolved separately in water at room temperature and always in nonmetallic containers; they recommend using one part of broth plus one part of water (50:50) and using it immediately. The broth should not be applied to very small, newly germinated or flowering seedlings [41,42]. It is important to note that most *quinoa* production in Peru is carried out under the conventional production system that promotes intensive soil use, the planting of commercial varieties, inorganic fertilizers and the control of diseases and pests with synthetic pesticides [29]. This is because short-term profitability is the main concern, but the indiscriminate use of agrochemicals deteriorates crop fields and damages farmers' health [32]. Evidently, the green revolution, which focused on germplasm breeding and the use of agrochemicals, contributed to the increase in food plant production. However, the ecological and social damage of an agriculture dependent on external inputs, particularly imported products, raises concerns. Most of the technical advice comes through the agrochemical stores and sellers, who, thanks to their entrepreneurial capacity, have taken advantage of the situation to promote agriculture dependent on marketable products such as chemical fertilizers and pesticides, which now account for half of the total cost of production. This has contributed to an overuse of toxic chemicals that are damaging ecosystems, which in turn has resulted in new outbreaks of pests and diseases, soils with diminishing biotic life and polluted waters. In addition, pests become more resistant to agrochemicals, which has led to a vicious cycle in which farmers need to apply more and more toxic chemicals each year, while the productivity of their land continues to decline [36,37].

Regarding the commercialization of *quinoa* and *achita* in Peru, there are companies that grow and stockpile these Andean grains for export. These companies pay a low price to the small farmer, which makes a big difference compared to the price of packaged products in shopping centers. In recent years, *quinoa* and *achita* exports have increased, but this development opportunity must be framed within a framework of fair and sustainable trade for the Andean population. Peru has signed and ratified the Nagoya Protocol on access to genetic resources and fair and equitable sharing of the benefits arising from their utilization; therefore, in accordance with this international treaty, fair and equitable sharing of the benefits arising from the utilization of these genetic resources must be achieved, and payment for these harvests must be adequate [43].

The medicinal uses of *quinoa* and *achita* have been known to Andean people since ancient times. In traditional indigenous medicine, the leaves, stems and grains were used as healing, anti-inflammatory and analgesic agents, and they were therefore frequently used to treat bone fractures, bruises and internal hemorrhages. They were also used as antiseptics for the urinary tract and to treat blennorrhoea and urinary complaints [44]. Its

lithium content reduces melancholy and sadness, phytoestrogens (daidzein and kinestein), can prevent uterine cancer and reduce menopausal problems as well as increase milk secretion. Its high calcium content, which is easily absorbed, prevents osteoporosis, and flavonoids promote blood circulation and reduce the risk of thrombosis. Anti-diabetic, antioxidant, anti-inflammatory, immunoregulatory, antimicrobial, anti-obesity, and heart-healthy properties are also being studied [4,10,11]. This scientific evidence supports the medicinal uses of *quinoa* and reveals that the use of its grains is not only good nutritionally but can also be used to improve health. This ancestral knowledge about the medicinal uses of *quinoa* and *achita* has been greatly eroded, as the results show that only 13.5% of the interviewed villagers remember any medicinal use of quinoa. The most common uses reported in our interviews are as a purgative (76.9%) and to relieve colic (23%). The use of quinoa as a *purgative* is not reflected in the works cited and, due to the high fiber content of these grains and leaves, is perfectly justified [45]. Although traditional knowledge about the medicinal use of Andean grains has been lost, Table 6 shows that the informants still have traditional knowledge about other medicinal plants in these districts; according to the data obtained, 91.7% of the informants use medicinal plants.

Traditional knowledge resides mainly in older people, and despite the efforts of many countries to preserve their ancestral knowledge, there has been a significant loss of traditional knowledge passed down from parents to children about the uses of the plants around them, mainly due to globalization, migration for education and work, and a reduced interest of young people in this cultural heritage [45,46]. This aspect is related to what is shown in Figures 2 and 3 of the research because it is adults over 45 years of age who still retain traditional knowledge about the medicinal plants and use them to take care of their health. Furthermore, various socio-cultural aspects of the relationship between human beings and ancestral Andean crops in the districts of Quinua and Acos Vinchos, whose economy is traditionally based on agriculture, handicrafts and little livestock, indicate that traditional medicine derived from ancestral knowledge is still maintained. Figures 3 and 4 show that the use, wild collection and cultivation of medicinal plants is very present among the inhabitants of both Quinua and Acos Vinchos. This indicates that knowledge about the medicinal use of plants, their location in the field and how to cultivate them in the garden still survives, especially among informants over 45 years of age. In Acos Vinchos, it is mainly women over 60 years of age who still maintain the traditional knowledge of their medicinal use and play the role of seed guardians. These results are in line with the studies carried out by Alberti-Manzanares y Luzuriaga-Quichimbo [23,47].

Most of these medicinal plants are collected in the field (36.4%) and from the orchard (20.5%), a very low percentage of which are bought (6.3%). A total of 91.7% report that their family uses medicinal plants, and 96.9% indicate that they would recommend their use to others. These results coincide with those reported by the WHO. Medicinal plants constitute a valuable resource in the health systems of developing countries, although there are no accurate data to assess the extent of global use of these plants, WHO has estimated that more than 80% of the world's population routinely uses traditional medicine to meet their primary health care needs and has developed the strategy on traditional medicine in 2014–2023 to support member states to harness their potential contribution to people's health, well-being and health care and to promote their safe and effective use through regulation and research [45,48]. As shown in Table 6, 78.1% of the villagers interviewed in Quinua and Acos Vinchos go to a health center, use medicinal plants and visit healer to improve their ailments. Although access to synthetic drugs is becoming easier, price can be a problem.

Random screening methods continue to be preferred in the search for active compounds by the pharmaceutical industry, but in recent years, special attention has been given to the use of ethnobotanical information for plant screening in the search for bioactive active compounds [45,49]. Indigenous Andean quinoa crops have excellent potential as a source of health-promoting bioactive compounds such as phenolic acids, saponins, phytosteroids and phytosterols, and in particular flavonoids; their content in *quinoa* grains is

exceptionally high. *Bilberry* berries have been considered an excellent source of flavonoids, especially quercetin and myricetin, but when compared, the levels are 5 to 10 times lower than those found in quinoa grains. *Quinoa* seeds can be considered a very good source of flavonoids [50]. Efforts must be made to prevent the definitive loss of traditional knowledge on medicinal plants, not only to preserve this cultural heritage, but also to record information on certain useful species, which could be relevant for the development of new sources of medicines and other benefits for humanity while contributing to the protection of biodiversity [45,51].

## 5. Conclusions

The research shows that traditional knowledge about the medicinal use and crop management of these Andean grains has been lost, although they are still important in their diet. Two of the main foods consumed by the inhabitants of Quinua and Acos Vinchos are *quinoa* and *achita*. Some 58.3% of the informants consume them in their diet, and 66.7% grow them for their own consumption. The consumption of these grains in their daily diet helps to reduce the nutritional problems of the rural population because they contain proteins of high biological value, as well as carbohydrates, polyunsaturated fatty acids, fiber, vitamins and minerals. In the districts studied, traditional knowledge of the medicinal use of these Andean grains has been lost, but knowledge of other medicinal plants in the area is still preserved.

The cultivation of *quinoa* and *achita* is a common activity and is considered a good alternative in the region, despite pests and drought and little technological support. This study shows that a very high percentage of the farmers interviewed (93.8%) use their own seeds for replanting, thus becoming independent from commercial seeds and maintaining local varieties. The loss of traditional techniques of soil preparation, use of organic matter and preparation of pest control products indicate a change in crop management. A total of 85.4% use agrochemicals, on which they depend, thus decreasing the productivity of their land and edaphic resilience, causing various types of negative impacts on the environment and, in some cases, harming their own health. The use of agrochemicals does not result in higher production for most farmers, as only 39.6% of the informants' report that their production is good.

Finally, we suggest a framework of intervention to reward and promote the use of seeds of local varieties, in order to inform about the negative consequences of the use of agrochemicals and to offer them alternatives that promote responsible and sustainable management of the resources used in production, starting with soil management: use of organic matter, application of good agricultural practices and use of biocides to control diseases and pests. All of this will help us to take up the knowledge that reconciles us with our natural environment with the help of today's technology.

**Author Contributions:** Conceptualization, R.B.A. and L.M.M.-C.; methodology, L.M.M.-C. and R.C.; software, R.C. (Reynán Cóndor) and E.D.L.C.; validation, L.M.M.-C. and R.C. (Reynán Cóndor); formal analysis, R.C. (Reynán Cóndor) and L.M.M.-C.; Investigation: R.B.A., R.C. (Roxana Carhuaz) and R.L.; resources, R.B.A. and R.C. (Reynán Cóndor); data curation, R.C. (Reynán Cóndor) and L.M.M.-C.; writing—original draft preparation, R.B.A., L.M.M.-C., E.D.L.C. and R.C. (Reynán Cóndor); writing—review and editing, R.B.A. and L.M.M.-C.; visualization, L.M.M.-C. and R.B.A.; supervision, R.B.A. and L.M.M.-C.; project administration, R.B.A. and L.M.M.-C.; funding Acquisition, R.B.A. All authors have read and agreed to the published version of the manuscript.

**Funding:** Part of the research work has been funded by the Vice-rectorate of Research of the National University of San Cristobal de Huamanga.

**Institutional Review Board Statement:** Not applicable.

**Informed Consent Statement:** Informed consent was obtained from all subjects involved in the study.

**Data Availability Statement:** Not applicable.

**Acknowledgments:** To Ranulfo Cavero Carrasco in the Research of the UNSCH for the financial support, to the inhabitants of the intervened districts, to Manuel Pardo for validating the survey, to Blga. Laura Aucasime Medina, responsible for the Herbarium Huamangensis of the Faculty of Biological Sciences of the National University of San Cristóbal de Huamanga (Ayacucho-Peru) and to the Bachelor in Biological Sciences Carolina Fernández for collaborating in the application of the surveys.

**Conflicts of Interest:** The authors declare that they have no conflict of interest.

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
