# Peer review of "Food and Medicinal Uses of Ancestral Andean Grains in the Districts of Quinua and Acos Vinchos (Ayacucho-Peru)"

_agronomy, doi:10.3390/agronomy12051014_

Round 1
Reviewer 1 Report
Dear author/s,
the topic of the paper: "Food and medicinal uses of ancestral Andean grains in the districts of Quinua and Acos Vinchos (Ayacucho-Peru). " is an interesting one, and the results could be useful for future development plans. However there are some aspect that should be improved:
- The aim of the paper is too general. The specific objectives and research questions/research hypotheses are missing. Please add them. At the same time organize the results and discussions section in accordance with the objectives of the research.
- However the statistics is basic, due to the topic the results might be interesting.
- Is not clear what gap the literature the current research fills. Which are the managerial implications of the study?
Please clarify the above mentioned aspects in order to increase the quality of the paper.
Author Response
Response to Reviewer 1 Comments
Dear reviewer
Thank you very much for your comments which have helped us to improve the quality of our paper.
- The aim of the paper is too general. The specific objectives and research questions/research hypotheses are missing. Please add them. At the same time organize the results and discussions section in accordance with the objectives of the research.
We have indicated at the end of the introduction section our research hypothesis, specifying the general objective and listing the specific objectives:
“Our research hypothesis is to study whether traditional knowledge on the use and management of Andean grains is still valid or has been lost due to globalization. The main aim of this research is to recover, evaluate and enhance local knowledge about ancestral Andean grains in the districts of Quinua and Acos Vinchos (Peru). To this end, the specific objectives of this study are: a. to collect data on the socio-economic characteristics of the informants; b. to evaluate the information on the nutritional use of these grains; c. to determine the type of management of these crops; d. to collect traditional knowledge on the medicinal use of these grains and assess its validity”
We have also reorganized the results and discussion section in accordance with the objectives.
- However, the statistics is basic, due to the topic the results might be interesting.
Indeed, we have discussed the results and drawn interesting conclusions. However, some tables and figures have been reorganised for a better understanding.
- Is not clear what gap the literature the current research fills. Which are the managerial implications of the study?
In one of the areas where quinoa production is among the highest in the world, the information provided by informants on the use and management of their Andean grain crops is important for management to act appropriately. This study shows that a very high percentage of the farmers interviewed (93.8%) use their own seeds for replanting, thus becoming independent from commercial seeds and maintaining local varieties. The loss of traditional techniques of soil preparation, use of organic matter and preparation of pest control products indicate a change in crop management. 85.4% use agrochemicals, on which they depend, thus decreasing the productivity of their land and edaphic resilience, causing various types of negative impacts on the environment and in some cases, harming their own health. The use of agrochemicals does not result in higher production for most farmers, as only 39.6% of the informants report that their production is good.
We have suggested a framework of intervention to reward and promote the use of seeds of local varieties, to inform about the negative consequences of the use of agrochemicals and to offer them alternatives that promote responsible and sustainable management of the resources used in production, starting with soil management: use of organic matter, application of good agricultural practices and use of biocides to control diseases and pests.
Kind regards
Reviewer 2 Report
This study evaluated the traditional knowledge regarding crop management, food, and medicinal uses of Andean grains amongst the inhabitants of districts of Quinua and Acos Vinchos (Ayacucho-Peru).
Lines 23–24 – The Latin names of the plants used as keywords are not mentioned in the manuscript. Please do so.
Tables 1 and 2 – Please use the impersonal expression of the syntagm “place where you live” (i.e. place of living, living region or even living place).
Table 2 – Please replace “anos” with years in the “Years of residence” part.
Table 3 – Please update the title so it reflects all the contents of the table (you should add the production of grains too, instead of only their usage).
Table 3 – I advise the authors not to use the questions as such while providing the results (i.e. instead of “What kind of ancient Andean grains do you use for food?” you can use “types of Andean grains used as food” and so on), thus providing the readers with a better understanding of the interpretation of the study. This remark stands for all the tables in the manuscript.
Table 6 – I find the stating of the places to go while sick alone and combined to be hard to read and understand. A simpler approach based only on the main places I believe would provide a better understanding of the frequency in which they are visited by people in need (i.e. Health center for Quinua = 30 people, Pharmacy for Quinua = 4 people, and so on). If you want to emphasize the way people combine these places, I suggest you do it in the text.
Figure 2 – I advise the authors to delete the question from the top of the figure, providing that they mentioned the information in the title of the figure.
Figures 3 and 4 – I advise the authors to delete the “Bar chart” and the question stated in these figures. The same remarks as for Table 6 regarding the interpretation of the data are standing for these figures too.
Figure 5 – I advise the authors to delete the “Bar chart” and the question stated in these figures.
Line 317 – Please provide the full name of MINAGRI the first time it appears in the text.
Lines 484–501 – The conclusions should reflect the results you obtained via the study you conducted (i.e. in the conclusions section you mention profitability and export, which are not found in your results). Please modify this section to meet this requirements.
Many of the references that you provided are old. Please search the literature for articles published within the last five years.
Author Response
Response to Reviewer 2 Comments
Dear reviewer
Thank you very much for your comments which have helped us to improve the quality of our paper.
Lines 23–24 – The Latin names of the plants used as keywords are not mentioned in the manuscript. Please do so.
We have mentioned the Latin names used as keywords in the manuscript.
Tables 1 and 2 – Please use the impersonal expression of the syntagm “place where you live” (i.e. place of living, living region or even living place).
We have made the changes
Table 2 – Please replace “anos” with years in the “Years of residence” part.
We have corrected the error
Table 3 – Please update the title so it reflects all the contents of the table (you should add the production of grains too, instead of only their usage).
We have added production
Table 3 – I advise the authors not to use the questions as such while providing the results (i.e. instead of “What kind of ancient Andean grains do you use for food?” you can use “types of Andean grains used as food” and so on), thus providing the readers with a better understanding of the interpretation of the study. This remark stands for all the tables in the manuscript.
We have made the changes
Table 6 – I find the stating of the places to go while sick alone and combined to be hard to read and understand. A simpler approach based only on the main places I believe would provide a better understanding of the frequency in which they are visited by people in need (i.e. Health center for Quinua = 30 people, Pharmacy for Quinua = 4 people, and so on). If you want to emphasize the way people combine these places, I suggest you do it in the text.
The table has been simplified for better understanding and we have clarified the explanation in the text
Figure 2 – I advise the authors to delete the question from the top of the figure, providing that they mentioned the information in the title of the figure.
We have removed the question from the top of the figure 2.
Figures 3 and 4 – I advise the authors to delete the “Bar chart” and the question stated in these figures. The same remarks as for Table 6 regarding the interpretation of the data are standing for these figures too.
We have simplified figure 3 in accordance with Table 6 and kept it as it provides information on the age of respondents who do not go to the health center and take care of their health by using medicinal plants and visiting the healer; interesting information to see who retains traditional knowledge about the medicinal use of plants and among what age range these practices are common.
We have eliminated figure 4 by including the most relevant information in the text.
Figure 5 – I advise the authors to delete the “Bar chart” and the question stated in these figures.
We have retained this figure because, together with figure 3 show that the use, wild collection and cultivation of medicinal plants is very present among the inhabitants of both Quinua and Acos Vinchos. This indicates that knowledge about the medicinal use of plants, their location in the field and how to cultivate them in the garden still survives, especially among informants over 45 years of age.
Line 317 – Please provide the full name of MINAGRI the first time it appears in the text.
We have corrected the error
Lines 484–501 – The conclusions should reflect the results you obtained via the study you conducted (i.e. in the conclusions section you mention profitability and export, which are not found in your results). Please modify this section to meet this requirements.
We have deleted that paragraph and modified this section to meet the requirements.
Many of the references that you provided are old. Please search the literature for articles published within the last five years.
We have revised the entire text and added information included in the updated bibliographical references that we have consulted and added to the bibliography. Also, we have revised the English language and style.
Kind regards
Reviewer 3 Report
- The discussion details and figure 1 should move to the discussion part. Method should be concise.
The authors worked well on this manuscript.
Author Response
Response to Reviewer 3 Comments
Dear reviewer
Thank you very much for your comments which have helped us to improve the quality of our paper.
- The discussion details and figure 1 should move to the discussion part. Method should be concise.
The last paragraph of this section has been moved to the discussion, but figure 1 has been kept in this section as it illustrates the study area.
We have removed redundant paragraphs in the Methodology section to make it clearer and more concise.
We have also revised the English language and style.
Kind regards
Round 2
Reviewer 2 Report
This study evaluated the traditional knowledge regarding crop management, food, and medicinal uses of Andean grains amongst the inhabitants of districts of Quinua and Acos Vinchos (Ayacucho-Peru).
Figure 2 – As I advised the authors before, please delete the question from the top of the figure (“¿Uses quinoa and/or cañihua as a medicinal plant?”), providing that you mentioned the information in the title of the figure.
Figure 4 (a and b)– As I advised the authors before, please delete the text “Bar chart” from the top of the figure (I did not advise the authors to delete the bar charts) and the question provided at the top of the legend (“How do you obtain medicinal plants?”).
Author Response
Dear reviewer
Apologies for not deleting in the previous manuscript what you suggested regarding figures 2 and 4.
Figure 2 – As I advised the authors before, please delete the question from the top of the figure (“¿Uses quinoa and/or cañihua as a medicinal plant?”), providing that you mentioned the information in the title of the figure.
We have deleted the question from the top of the figure 2
Figure 4 (a and b)– As I advised the authors before, please delete the text “Bar chart” from the top of the figure (I did not advise the authors to delete the bar charts) and the question provided at the top of the legend (“How do you obtain medicinal plants?”).
We have deleted the text “Bar chart” and the question from the figure 4
Kind regards